# Q-Sparse: All Large Language Models can be Fully Sparsely-Activated

## Abstract

We introduce, **Q-Sparse**, a simple yet effective approach to training sparsely-activated large language models (LLMs). Q-Sparse enables **full sparsity of activations** in LLMs which can bring significant efficiency gains in inference. This is achieved by applying top-$K$ sparsification to the activations and the straight-through-estimator to the training. We also introduce **Block Q-Sparse** for batch training and inference. The key results from this work are, (1) Q-Sparse can achieve results comparable to those of baseline LLMs while being much more efficient at inference time; (2) We present an inference-optimal scaling law for sparsely-activated LLMs; (3) Q-Sparse is effective in different settings, including training-from-scratch, continue-training of off-the-shelf LLMs, and finetuning; (4) Q-Sparse works for both full-precision and 1-bit LLMs (e.g., **BitNet b1.58** (Wang et al., 2023)). Particularly, the synergy of BitNet b1.58 and Q-Sparse (can be equipped with MoE) provides the cornerstone and a clear path to revolutionize the efficiency, including cost and energy consumption, of future LLMs.

## 1 Introduction

Large language models (LLMs) have achieved remarkable performance on a wide range of natural language processing (NLP) tasks. However, the deployment of LLMs in real-world applications is challenging due to their high computational cost and memory footprint, especially during the inference stage. To address this challenge, recent works (Ma et al., 2024; Wang et al., 2023; Song et al., 2024b; Xia et al., 2023; Leviathan et al., 2023) have focused on improving the efficiency of LLMs with various approaches, including quantization (Ma et al., 2024; Wang et al., 2023; Frantar et al., 2023), pruning (Xia et al., 2023), distillation (Gu et al., 2023), better decoding (Leviathan et al., 2023), and so on. One promising approach is to use sparsity to reduce the number of activated parameters in LLMs.

Sparsity contributes two factors to the efficiency of LLMs. First, sparsity can reduce the amount of computation of the matrix multiplication as zero elements are not computed. Second, sparsity can reduce the amount of input/output (I/O) that transfers the parameters between the memory and the computation units. The I/O transfer serves as the major bottleneck in the inference stage of LLMs.

One common approach to sparsity in LLMs is to use weight sparsity, which prunes the model weights to save the computation. However, unstructured weight sparsity is difficult to parallelize in GPU devices, while structured weight sparsity has a large impact to the accuracy of the model.

Another approach is to use activation sparsity, which reduces the number of activated elements in the activation tensors. Activation sparsity can be achieved by using the mixture-of-experts (MoE) mechanism (Lepikhin et al., 2021; Fedus et al., 2021), modifying the activation function (Mirzadeh et al., 2023; Song et al., 2024b), or predicting the position to be sparsed (Liu et al., 2023). However, these approaches do not enable full sparsity of activations in LLMs, which can limit the efficiency gains during the inference stage. Moreover, compared to the dense models, the scaling laws for the sparsely-activated LLMs have not been well studied.

To explore the full potential of sparsity in LLMs, we introduce **Q-Sparse**, a simple yet effective approach to enable full sparsity of activations in LLMs. The major modification on LLMs is in the linear projection (i.e., matrix multiplication). As shown in Figure 1, for each linear projection, it has a top-K sparsification function that selects the top-K activations in the input tensor. For the

backprogation, we use the straight through estimator to compute the gradients of the activations. We also introduce a squared ReLU function for the feed-forward layers to further improve the sparsity of the activations. Q-Sparse can be used with both full-precision and quantized LLMs. Furthermore, we introduce, **Block Q-Sparse**, a block sparsity implementation to make Q-Sparse compatible with batch training and inference.

To study the scaling law of sparsely-activated LLMs, we conduct a series of scaling experiments and derive an inference-optimal scaling law for sparsely-activated LLMs. We summarize the findings from the scaling experiments and the implications of the scaling law as below:

- The performance of the sparsely-activated models is better than the dense baselines with the same inference compute budget (i.e., activated parameters or FLOPs).
- As the parameters $N$ scales, the performance gap between the sparsely-activated models and the dense baselines decreases.
- The performance of the sparsely-activated models with around 40% sparsity ratio can match the performance of the dense baselines with the same model size and training tokens.
- Given the same inference budget $N_a$, a sparsely-activated full-precision model with a sparsity ratio of 45.58% (or $1.84N_a$ parameters) can achieve the best performance. For the 1.58-bit models, the optimal sparsity ratio is 61.25%.

We also conduct experiments to evaluate the effectiveness of Q-Sparse in different settings, including training-from-scratch, continue-training of off-the-shelf LLMs, and finetuning. We show that Q-Sparse can achieve results comparable to those of baseline LLMs with the same training cost while being much more efficient at inference time.

## 2 Q-SPARSE: FULLY SPARSELY-ACTIVATED LLMS

### 2.1 ARCHITECTURE

The Q-Sparse architecture is based on the Transformer architecture (Vaswani et al., 2017; Touvron et al., 2023) with modifications to enable sparsity in the activations.

**Top-K Sparsity**

The Transformer architecture uses *nn.Linear* to perform the projection in both attention and feed-forward layers, which can be written as:

$$\mathbf{Y} = \mathbf{X} \cdot \mathbf{W}^T \tag{1}$$

where $\mathbf{X} \in \mathbb{R}^{N \times D}$ is the input tensor, $\mathbf{W} \in \mathbb{R}^{M \times D}$ is the weight tensor, and $\mathbf{Y} \in \mathbb{R}^{N \times M}$ is the output tensor. The *nn.Linear* operation is equivalent to the matrix multiplication operation.

We introduce a top-K sparsity function on top of the matrix multiplication operation. The top-K sparsity function is defined as:

$$\mathbf{Y} = (\mathbf{X} \odot \mathbf{M}) \cdot \mathbf{W}^T, \ \mathbf{M} = \text{Top}_k(|\mathbf{X}|) \tag{2}$$

where $\mathbf{M} \in \mathbb{R}^{N \times D}$ is the mask tensor that indicates the top-K activations in the input tensor $\mathbf{X}$ in terms of the absolute values, $\odot$ is the element-wise multiplication operation, and $\text{Top}_k$ is the function that selects the top-K elements in the tensors. To reduce the interval around zero, we re-scale the tensor by its $L_2$ norm after performing the top-K sparsity function.

**Quantized Top-K Sparsity**

Recent works (Wang et al., 2023) have shown that quantization can be used to reduce the memory footprint and computational cost of LLMs without the loss of performance. We introduce a quantized version of the top-K sparsity function. The quantized top-K sparsity function is defined as:

$$\mathbf{Y} = (\text{Q}(\mathbf{X}) \odot \mathbf{M}) \cdot \mathbf{W}^T \tag{3}$$

where $\text{Q}(\cdot)$ is the quantization function that quantizes the input tensor $\mathbf{X}$ to a 8-bit representation:

$$\text{Q}(X) = \text{RoundClip}(\frac{127}{\gamma + \epsilon}\mathbf{X}, -128, 127), \ \gamma = \max(|\mathbf{X}|) \tag{4}$$

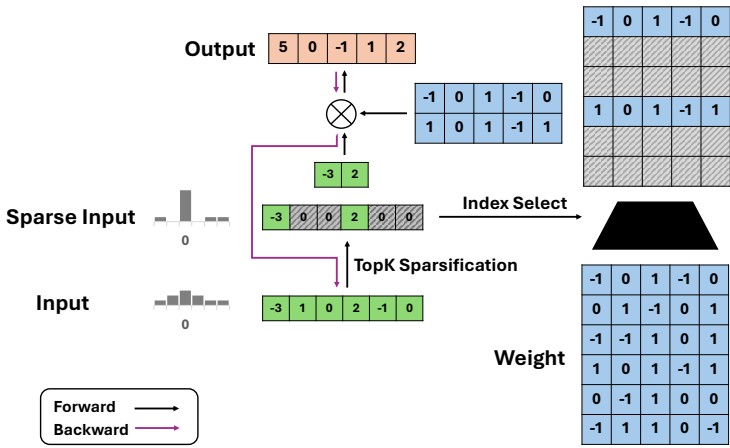

Figure 1: Q-Sparse achieves a superior inference-optimal scaling law than the dense models. It saves significant compute of matrix multiplication by top-$K$ sparsification of the activations.

$$\text{RoundClip}(X, a, b) = \min(\max(\text{round}(X), a), b) \tag{5}$$

where $\epsilon$ is a small constant to avoid division by zero, and $\gamma$ is the maximum absolute value in the input tensor $\mathbf{X}$.

Q-Sparse can be used with both full-precision and quantized LLMs. Specifically, the quantized version of Q-Sparse is compatible with 1-bit LLMs, such as BitNet b1.58 (Wang et al., 2023). When using Q-Sparse with 1-bit LLMs, the quantization function is performed on the weight tensor $\mathbf{W}$:

$$\mathbf{Y} = (\mathbf{Q}(\mathbf{X}) \odot \mathbf{M}) \cdot \mathbf{Q}_w(\mathbf{W})^T \tag{6}$$

where $\mathbf{Q}_w(\cdot)$ is the absmean function that quantizes the weight tensor $\mathbf{W}$ to a 1.58-bit representation:

$$\mathbf{Q}_w(W) = \text{RoundClip}(\frac{\mathbf{W}}{\alpha + \epsilon}, -1, 1), \ \alpha = \text{mean}(|\mathbf{W}|) \tag{7}$$

where $\alpha$ is the mean absolute value in the weight tensor $\mathbf{W}$.

**Squared ReLU**

To further improve the sparsity of the activations, we use the squared ReLU function (So et al., 2021) for the feed-forward layers. The squared ReLU function is defined as $\text{ReLU}(\mathbf{X})^2$. Following the LLaMA architecture, we use the gated linear unit (GLU) for the feed-forward layers. The squared ReLU function is applied with the GLU function into a $\text{ReLU}^2\text{GLU}$ function. It is defined as:

$$\text{ReLU}^2\text{GLU}(\mathbf{X}) = \mathbf{X}\mathbf{W}_{\text{up}}^T \odot \text{ReLU}^2(\mathbf{X}\mathbf{W}_{\text{gate}}^T) \tag{8}$$

**Block Q-Sparse**

While the top-K sparsification can be used in the single-sample mode, it is not friendly with the batch mode for the current GPU devices. Recent work (Zhou et al., 2021; Lin et al., 2023) shows that N:M sparsity, where N out of M consecutive elements to be zero, is more hardware friendly and can be used in the batch mode with an optimized GPU kernel. To leverage this feature of the modern GPU devices, we introduce Block Q-Sparse. The key idea of Block Q-Sparse is to apply the top-K sparsity function on the activations in the block level, and the block size is set to $M$ so that there are always $M - K$ zeros out of $M$ consecutive values. The top-K sparsity function is applied to the activations in each block independently. The block level sparsity can be used to reduce the memory footprint and computational cost of the LLMs in the batch mode.

## 2.2 TRAINING

Most of the existing works (Mirzadeh et al., 2023) on training sparsely-activated models use the vanilla back-propagation algorithm to compute the gradient through the sparsity function:

$$\frac{\partial \mathbf{Y}}{\partial \mathbf{X}} = \frac{\partial \mathbf{Y}}{\partial (\mathbf{X} \odot \mathbf{M})} \odot \mathbf{M} \tag{9}$$

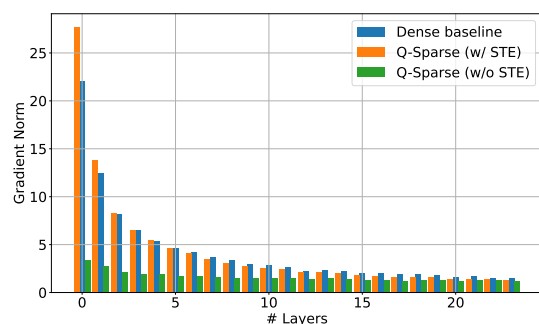

Figure 2: The average magnitude of each projection's gradient of dense baseline, Q-Sparse with and without STE across different layers. The visualization is conducted with 300M model size on a subset of the valid set of C4 (Raffel et al., 2019). It shows that the gradient vanishes without STE.

where $\mathbf{M}$ is the mask tensor that indicates the top-K activations in the input tensor $\mathbf{X}$, and $\odot$ is the element-wise multiplication operation.

The vanilla back-propagation algorithm has a limitation. It zero-outs the gradients of the non-activated elements, which can lead to the vanishing gradient problem, especially when the sparsity ratio is high. In this work, we propose to use the straight-through estimator (Bengio et al., 2013) to back-propagate the gradients through the sparsity function. In this way, the gradients are passed through the sparsity function without being zeroed-out. The straight-through estimator is defined as:

$$\frac{\partial \mathbf{Y}}{\partial \mathbf{X}} = \frac{\partial \mathbf{Y}}{\partial (\mathbf{X} \odot \mathbf{M})} \tag{10}$$

We visualize the average $l2$ norm of each projection's gradient across different layers for dense model, Q-Sparse with and without STE. We adopt top-K as 50% for Q-Sparse. Without STE, the gradient is much smaller at the bottom layers, while STE can preserve the magnitude of the gradients. As shown in Figure 2, STE estimator significantly eases the issue of gradient vanishing, especially at the bottom of the layers. We present more visualizations for each components in the Figure 8 of Appendix A.

### 2.3 Q-Sparse for Continue-Train and Finetuning Settings

Q-Sparse can be used in different settings, including training-from-scratch, continue-training, and finetuning. In the continue-train and finetuning settings, we use the same architecture and training procedure as in the training-from-scratch setting. The only difference is that we initialize the model with the pre-trained weights and continue training with the sparsity function enabled.

For the pre-trained models that do not have the squared ReLU function in the feed-forward layers, we apply the top-K sparsity function after the activated function (e.g., SiLU) in the feed-forward layers. It can improve the sparsity of the activations without changing the model architecture.

## 3 Scaling Laws

Recent work on large language models has shown that the performance of LLMs scales with the model size and the amount of training data. Hoffmann et al. (2022) argues that the converged performance of a dense Transformer model with $N$ parameters follows a power-law scaling law, which can be written as:

$$L(N) \triangleq E + \frac{A}{N^\alpha} \tag{11}$$

where $L(N)$ is the performance of the model with $N$ parameters, $E$ is the performance of the model with infinite parameters, $A$ is a constant, and $\alpha$ is the scaling exponent. Note that the number of training tokens are fixed in this setting, which is part of the constant $E$.

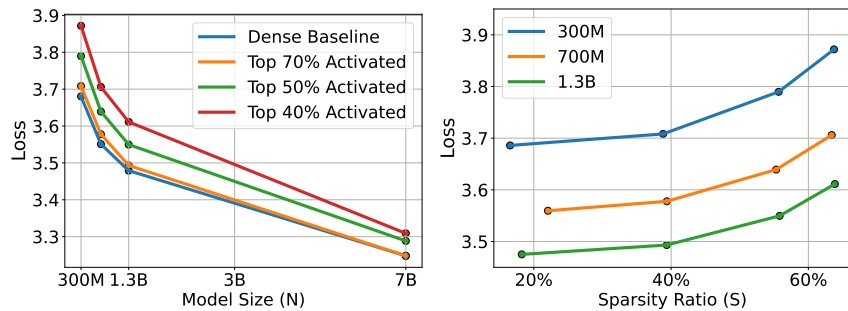

Figure 3: The scaling curves of the sparsely-activated models regrading to the model size given a fixed sparsity ratio $S$ (Left), and regrading to the sparsity ratio given a fixed model size $N$ (Right).

In this work, we investigate the scaling law of sparsely-activated LLMs. We find that the performance of sparsely-activated LLMs also follows a power-law scaling law, which can be written as:

$$L(N, S) \triangleq E + \frac{A(S)}{N^\alpha} \tag{12}$$

$$A(S) = B + C \exp\left(\frac{\beta}{1 - S}\right) \tag{13}$$

where $L(N, S)$ is the performance of the sparsely-activated model with $N$ parameters and a sparsity ratio of $S$, and $\alpha$ and $\beta$ are the scaling exponents.

In the following part, we will introduce how we derive the scaling law and the corresponding findings.

### 3.1 SCALING EXPERIMENTS AND FINDINGS

To determine the form of the scaling law of sparse-activated LLMs, we begin with a series of scaling experiments. In the experiments, we train a series of language models with Q-Sparse of various scales, ranging from 300M to 7B. The models are trained on the Redpajama dataset (TogetherAI, 2023). We use the Sentencepiece tokenizer from LLaMA to preprocess data. Besides Q-Sparse, we also train the dense baselines with the same datasets and settings. More details can be found in the Appendix B.

The observed losses of the sparsely-activated models and the dense baselines are shown in Figure 3. We summarize the findings as below:

- The performance of the sparsely-activated models scales with the model size and the sparsity ratio.
- Given a fixed sparsity ratio $S$, the performance of the sparsely-activated models follows a power-law scaling law with regards to the model size $N$.
- Given a fixed parameters $N$, the performance of the sparsely-activated models follows an exponential-law scaling law with regards to the sparsity ratio $S$.
- As the parameters $N$ scales, the performance gap between the sparsely-activated models and the dense baselines decreases.

According to these findings, our main hypothesis is that the performance of the sparsely-activated models follows a combination of a power-law scaling law with regards to the model size $N$ and an exponential-law scaling law with regards to the sparsity ratio $S$.

### 3.2 POWER LAW IN THE MODEL SIZE $N$

With a fixed sparsity ratio $S$, the scaling law should follows Kaplan et al. (2020)'s scaling law, which can be written as:

$$L(N, S) \triangleq E + \frac{A(S)}{N^{\alpha(S)}} \tag{14}$$

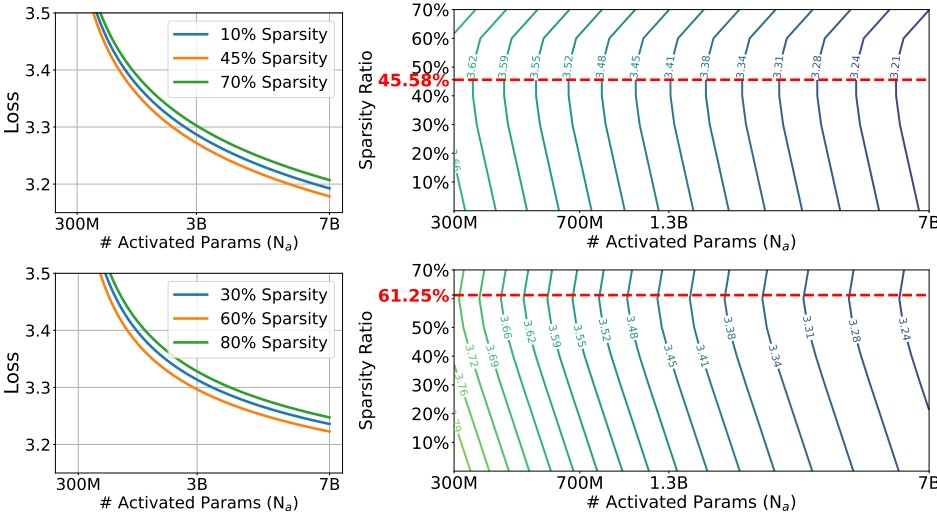

Figure 4: The inference-optimal scaling curves of the sparsely-activated models with full-precision (Top) and 1.58-bit (Bottom) weight. It shows that a sparisty of 45.58% for full-precision models and 61.25% for 1.58-bit models can achieve the best performance with the same inference compute budget (i.e., activated parameters or FLOPs).

where $\alpha(S)$ is the scaling exponent, and the scaling factor $A(S)$ is a function of the sparsity ratio $S$. Given any model size $N$, the function $L(N, S)$ should follow the Lipschitz continuity with regards to the sparsity ratio $S$. Therefore, the scaling exponent $\alpha(S)$ should be a non-decreasing function. Given any model size $N$, the function $L(N, S)$ is increasing with the sparsity ratio $S$, so $\alpha(S)$ should be a non-increasing function. Above all, the scaling exponent $\alpha(S)$ should be a constant, and the scaling function can be written as:

$$L(N, S) \triangleq E + \frac{A(S)}{N^\alpha} \tag{15}$$

## 3.3 Exponential Law in the Sparsity Ratio $S$

According to the above finding, the performance of the sparsely-activated models follows an exponential-law scaling law with regards to the sparsity ratio $S$. Therefore, the scaling factor $A(S)$ should also follow an exponential law. Besides, given any model size $N$, the scaling function is increasing with the sparsity ratio $S$. Therefore, the scaling factor $A(S)$ should be a non-decreasing function. The scaling factor $A(S)$ can be written as:

$$A(S) = B + C \exp\left(\frac{\beta}{1 - S}\right) \tag{16}$$

where $B$ is the scaling factor for extremely sparse LLMs, $C$ is the scaling factor for dense LLMs, and $\beta$ is the scaling exponent of the scaling factor $A(S)$ with regards to the sparsity ratio $S$.

## 3.4 Fitting the Parameters

We fit the parameters of the scaling law to the observed losses of the sparsely-activated models. We use the L-BFGS algorithm (Nocedal, 1980) to minimize the Huber loss (Huber, 1992) between the predicted and observed log loss.

$$\min_{E, B, C, \beta, \alpha} \sum_{\text{Runs } i} \text{Huber}_\delta \left(\log \hat{L}(N_i, S_i) - \log L_i\right) \tag{17}$$

Following Hoffmann et al. (2022), $\delta$ is set as $10^{-3}$. We select the best fit from a grid of initialisations around possible local optimas. $E$, $B$, $C$, $\alpha$ and $\beta$ are estimated as 1.86, 0.01, 1.89, 0.10 and 0.05, respectively.

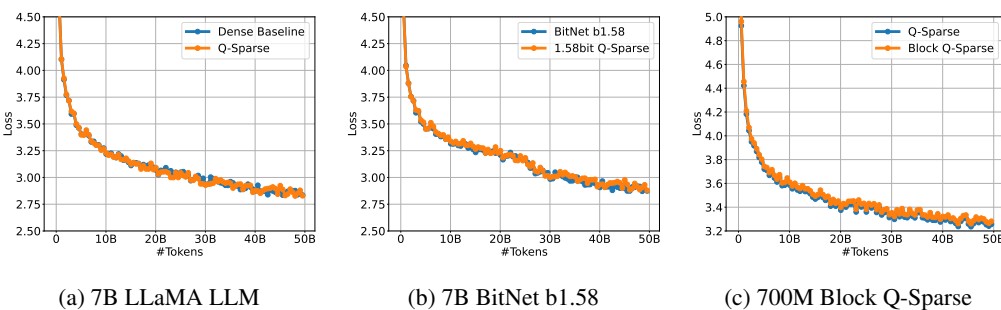

(a) 7B LLaMA LLM       (b) 7B BitNet b1.58       (c) 700M Block Q-Sparse

Figure 5: The training loss curve of Q-Sparse and the baseline with full-precision (a) and 1.58-bit (b) weight, Block Q-Sparse. We adopt top-$K$ as 70% for the experiments of BF16 and 1.58-bit weight, resulting in 40% overall sparsity. For the comparison with Block Q-Sparse, the sparsity ratio is 50% and the block size is set as 32.

### 3.5 DIMINISHING GAP BETWEEN SPARSELY-ACTIVATED MODELS AND DENSE BASELINES

Given the above scaling law, we can derive the performance of the sparsely-activated models and the dense baselines with the same model size $N$ and the same sparsity ratio $S$. The performance gap between the sparsely-activated models and the dense baselines decreases as the model size $N$ scales. The performance gap can be written as:

$$L(N, S) - L(N, 0) = \frac{A(S)}{N^{\alpha(S)}} - \frac{A(0)}{N^{\alpha(0)}} = \frac{A(0)}{N^{\alpha}}\left(\frac{A(S)}{A(0)} - 1\right) \tag{18}$$

Since $\alpha$ is a constant that satisfies $\alpha > 0$, the performance gap decreases as the model size $N$ scales. It means that given a large enough model size $N$, the performance of the sparsely-activated models can eventually match the performance of the dense baselines with the same model size.

### 3.6 INFERENCE-OPTIMAL SCALING LAW

The scaling law can also be transformed into a form that is dependent on the activated parameters $N_a$, which reflects the effective compute (i.e., FLOPs) of the model during inference:

$$L(N_a, S) \triangleq E + A(S)\left(\frac{1-S}{N_a}\right)^{\alpha} \tag{19}$$

where $N_a$ is the number of activated parameters in the model, which is equal to $N \times (1 - S)$. Since $A(S)$ is an increasing function and $(1 - S)^{\alpha}$ is a decreasing function, there exists a sparsity ratio $S^* > 0$ that minimizes the loss of the sparsely-activated models. This leads to the inference-optimal scaling law of the sparsely-activated models:

$$L(N_a) \triangleq E + A(S^*)\left(\frac{1-S^*}{N_a}\right)^{\alpha} \tag{20}$$

It shows that the performance of the sparsely-activated models is better than the dense baselines with the same inference compute budget $S^*$. We further solve the optimal sparsity ratio $S^*$, finding that $S^* \approx 45.58\%$. It means that a sparsely-activated model with a sparsity ratio of 45.58% (or $1.84N_a$ parameters) can achieve the best performance with the same inference budget $N_a$. We follow the same process to estimate the inference-optimal scaling law for 1.58-bit Q-Sparse models. We find that the optimal sparsity ratio is 61.25% (or $2.58N_a$ parameters). Figure 4 shows the inference-optimal scaling curves of the sparsely-activated models with full-precision and 1.58-bit weight. It shows that with the same performance, the sparsely-activated models can achieve a significant reduction in the number of activated parameters or FLOPs during inference.

The inference-optimal scaling law shows that the performance of the sparsely-activated models can be optimized by adjusting the sparsity ratio $S$. It can be used to guide the training of the sparsely-activated models and to optimize the performance of the models during inference.

## 4 EXPERIMENTS

We conduct experiments to evaluate the effectiveness of Q-Sparse in different settings, including training-from-scratch, continue-training of off-the-shelf LLMs, and finetuning.

### 4.1 TRAINING-FROM-SCRATCH

**Setting** We train a series of language models with Q-Sparse in both full-precision and 1.58 bits. The models are trained with 50B tokens on the Redpajama dataset (TogetherAI, 2023). We compare Q-Sparse with the dense baselines with the same datasets and settings.

**Results** The observed losses of the sparsely-activated models and the dense baselines are shown in Figure 5a. It shows that Q-Sparse with 40% sparsity ratio can match the performance of the dense baselines with the same model size and training tokens. The loss curves of 700M models are shown in Figure 7a of Appendix A.

**BitNet b1.58 + Q-Sparse** We further evaluate the effectiveness of Q-Sparse on 1-bit LLMs. We train a series of BitNet b1.58 models with Q-Sparse of various scales. We plot the training loss curves of both Q-Sparse and the BitNet b1.58 baseline. Figure 5b shows that the performance of the sparsely-activated BitNet b1.58 models is better than the dense baselines with the same inference compute budget. It demonstrates that Q-Sparse is compatible to 1-bit LLMs and their synergy can be used to optimize the performance of the models during inference. We present the loss curves of 700M models in Figure 7b of Appendix A.

**Block Q-Sparse** We evaluate the effectiveness of Block Q-Sparse. We compare it with Q-Sparse of the same sparsity ratio. The sparsity ratio is 50%, and the block size is set to 32 (i.e., N:M=16:32). The experiments are performed with the model sizes of 300M and 700M. The training loss curves of Q-Sparse and Block Q-Sparse are shown in Figure 5c. We present the loss curves of 300M models in Figure 7c of Appendix A. It shows that Block Q-Sparse has a similar convergence to Q-Sparse with the same sparsity. It demonstrates that Block Q-Sparse can match the performance of Q-Sparse when training from scratch.

**Ablation Study of top-K Sparisty and STE** To evaluate the effect of the top-K sparsity function, we compare the performance of the sparsely-activated models with the top-K sparsity function and the ReLU sparsity function. Moreover, we study the effect of the STE by comparing the models with and without STE. Figure 6 illustrates the results. It shows that either removing STE or replacing with ReLU function significantly hurt the performance. Besides, the sparsity ratio of the models with the ReLU function decreases as the training processes. In constrast, the sparisty ratio remains unchanged with the top-K sparisty function. As shown in Figure 9 of Appendix A, we break down the contribution of the sparsity ratio from different components, finding that the decreasing sparisty is mainly from the QKV projection, the gating projection and the up projection of the feed-forward layers. This proves the superior of top-K over ReLU function.

### 4.2 CONTINUE-TRAINING

**Setting** We continue-train the Mistral 7B model (Jiang et al., 2023) for 40B tokens on the FineWeb-Edu dataset (Lozhkov et al., 2024). We use the Sentencepiece tokenizer from Mistral to preprocess data. We use the batch size of 4M tokens and the learning rate of 5e-5. We use the Adam optimizer with the weight decay of 0.01. More training details can be found in Appendix B.

**Results** For a fair comparison, we continue-train the Mistral 7B model with the same recipe as the dense baseline. We compare Q-Sparse with the ReLUfication (Mirzadeh et al., 2023) and dReLU Sparsification (Song et al., 2024b) methods, which sparsify the model by changing the activation function. Following the origin paper (Mirzadeh et al., 2023), we adopt a two-stage training strategy that first replaces the non-ReLU activation and then adds the ReLU functions. For the dReLU Sparsification method, we implement the dReLU sparsification method following the origin paper (Song et al., 2024b). We evaluate these models on a range of language tasks using EleutherAI LM Harness (Gao et al., 2024), including 25-shot ARC-Challenge (Yadav et al.,

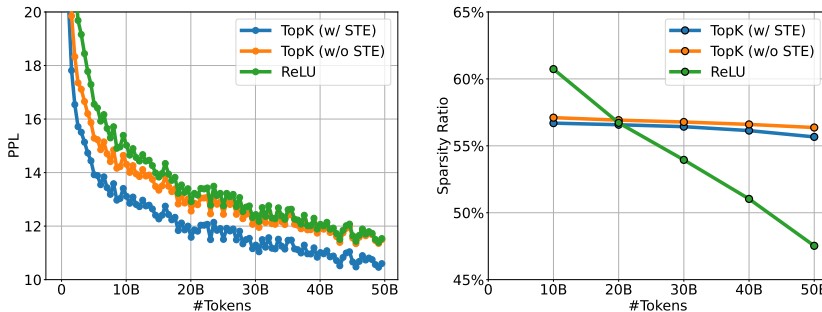

Figure 6: The training loss curves (Left) and the overall sparsity ratio (Right) of different sparsity functions. All models are trained with 300M size and 50B tokens.

Table 1: The results of the continue-training for Q-Sparse and the baselines on the end tasks.

| Models | Activated | ARC | HS | MMLU | WG | TQA | Avg. |
|---|---|---|---|---|---|---|---|
| Dense Baseline | 7.0B | 61.8 | 81.4 | 59.8 | 77.5 | 42.7 | 64.6 |
| ReLUfication (Mirzadeh et al., 2023) | 5.0B | 57.2 | 78.8 | 54.7 | 74.7 | 38.8 | 60.8 |
| dReLU Sparsification (Song et al., 2024b) | 5.4B | 59.2 | 78.0 | 54.0 | 75.8 | 38.3 | 61.0 |
| **Q-Sparse (this work)** | 2.9B | 59.0 | 79.0 | 55.6 | 74.0 | 41.0 | 61.7 |
| | 3.8B | 60.5 | 80.7 | 58.0 | 75.9 | 43.5 | 63.7 |

2019), 10-shot HellaSwag (Zellers et al., 2019), 5-shot Winogrande (Sakaguchi et al., 2020), 5-shot MMLU (Hendrycks et al., 2021) and zero-shot TruthfulQA (Lin et al., 2022). Results are shown in Table 1. It shows that Q-Sparse achieves comparable performance to the dense baseline while being much more efficient at inference time. Moreover, Q-Sparse outperforms the ReLUfication and dReLU Sparsification methods in terms of the performance and the sparsity ratio.

To break down the sparsity of each component in the model, we present the sparsity ratio of the query, key, value, output, up, down, and gate tensors in Table 3 of Appendix A. It shows that Q-Sparse achieves a higher sparsity ratio than the ReLUfication and dReLU Sparsification methods. The sparsity ratio of the query, key, value, output, up, and down tensors is higher than 40%, and the sparsity ratio of the gate tensor is higher than 60%. It demonstrates that Q-Sparse can achieve full sparsity of activations in LLMs.

## 4.3 SUPERVISED FINETUNING

**Setting** We finetune the base model of Mistral 7B (Jiang et al., 2023) and Qwen1.5 7B (Bai et al., 2023) on Open-Orca dataset (Lian et al., 2023) for both the dense baselines and Q-Sparse. The batch size is set as 128. The learning rates are selected from {3e-6, 5e-6, 7e-6}. All models are trained with 1 epoch for a fair comparison. The hyper-parameters are detailed in Appendix B. The evaluation is consistent with the experiments shown in Section 4.2.

**Results** The results are shown in Table 2. It shows that Q-Sparse with 3.6B activated parameters achieves significant better performance than the Qwen1.5 4B dense model. Moreover, Q-Sparse with around 4B activated parameters achieves comparable performance to the Mistral 7B model and the Qwen1.5 7B model. It demonstrates that Q-Sparse can be used to finetune a dense pretrained model to a much more efficient sparse model with almost no loss at accuracy.

## 4.4 EVALUATION OF BLOCK Q-SPARSE

**Setting** We finetune the base model of Mistral 7B (Jiang et al., 2023) and Qwen1.5 7B (Bai et al., 2023) on Open-Orca dataset (Lian et al., 2023) for Block Q-Sparse. The block size is set as 32,

Table 2: The results of the supervised fine-tuning for Q-Sparse, Block Q-Sparse and the dense baselines on the end tasks.

| Models | Activated | ARC | HS | MMLU | WG | TQA | Avg. |
|---|---|---|---|---|---|---|---|
| Qwen1.5-4B | 3.2B | 42.8 | 68.2 | 53.6 | 67.1 | 47.9 | 55.9 |
| Qwen1.5-7B | 6.5B | 47.7 | 74.6 | 61.5 | 71.4 | 50.7 | 61.2 |
| **Block Q-Sparse** | 3.6B | 47.0 | 71.1 | 56.7 | 67.6 | 50.5 | 58.6 |
| | 4.1B | 47.2 | 73.1 | 59.7 | 69.0 | 49.7 | 59.7 |
| **Q-Sparse** | 3.6B | 46.3 | 72.6 | 59.1 | 67.5 | 50.3 | 59.2 |
| | 4.1B | 47.9 | 73.2 | 59.2 | 69.4 | 51.1 | 60.1 |
| Mistral-7B | 7.0B | 62.5 | 82.6 | 61.2 | 77.6 | 50.3 | 66.8 |
| **Block Q-Sparse** | 3.8B | 59.7 | 80.6 | 58.7 | 75.5 | 50.3 | 65.0 |
| | 4.3B | 60.0 | 81.4 | 59.9 | 76.8 | 51.3 | 65.9 |
| **Q-Sparse** | 3.8B | 60.5 | 81.5 | 60.0 | 77.1 | 50.5 | 65.9 |
| | 4.3B | 61.4 | 81.6 | 60.6 | 77.6 | 50.7 | 66.4 |

which is recommended by the previous work (Lin et al., 2023) on N:M sparse kernels. The other hyper-parameters are consistent with the experiments shown in Section 4.3.

**Results**   Table 2 summarizes the results for Block Q-Sparse. Similar to the results of Q-Sparse, Block Q-Sparse achieves comparable performance to the dense baselines with much fewer activated parameters. It demonstrates that Block Q-Sparse can be used for a much more efficient sparse model while supporting the batch mode.

# 5 RELATED WORK

The magnitude of the inputs to the linear projections in LLMs often follow a long-tailed distribution, thus activation sparsity is a natural approach to reduce the inference cost while maintaining competitive performance. Liu et al. (2023) showed that the activation sparsity exists, can be predicted with low-cost algorithms. Mirzadeh et al. (2023) demonstrated that compared with widely-adopted SiLU function, using ReLU function has a negligible impact on convergence and performance while reducing computation and weight transfer. They further inserted the ReLU function before each linear projection to boost the overall sparsity of LLMs. PowerInfer (Song et al., 2023) uses the sparsity of down projection in feed-forward layers to design a GPU-CPU hybrid inference engine: hot-activated neurons are preloaded onto the GPU, while cold neurons are computed on the CPU. It reduces GPU memory demands and CPU-GPU data transfers. TurboSparse (Song et al., 2024b) proposed dReLU activation function to further improve the performance and activation sparsity. ProSparse (Song et al., 2024a) adopted progressive sparsity regularization to smoothly increase the sparsity, which mitigates performance degradation from radical shifts in activation distributions.

# 6 CONCLUSION AND FUTURE WORK

We introduce Q-Sparse, a simple yet effective approach to enable full activation sparsity in LLMs. Q-Sparse can achieve comparable performance to dense LLMs while being much more efficient at inference time. We present an inference-optimal scaling law for sparsely-activated LLMs. Q-Sparse is effective in different settings, including pre-training, continue-training and fine-tuning. More importantly, Q-Sparse is orthogonal and can be seamlessly integrated with MoE, and works for 1-bit LLMs (e.g., BitNet b1.58 (Wang et al., 2023)).

Recent works (Song et al., 2023; Liu et al., 2024) have shown promising end-to-end speedup with activation sparsity. The custom kernel for Q-Sparse can be easily implemented since Q-Sparse adopts the token-level sparsity. We leave this as a part of future work. In addition, we would like to scale up the training of 1-bit LLMs (i.e., BitNet b1.58) with fully sparse activations (i.e., Q-Sparse) in terms of both model size and training tokens. Furthermore, we will incorporate YOCO (Sun et al., 2024) to address the issue of KV cache for LLM inference.

## 7 ETHICS STATEMENT

In this work, we explore to use activation sparsity to improve the efficiency of large language models. Like most of the existing pre-trained models, our method may have some potential bias originating from the pre-training data.

## 8 REPRODUCIBILITY STATEMENT

Q-Sparse is simple and can be easily implemented for the existing architecture of large language models. We present details about Q-Sparse in Section 2, including top-$K$ sparsification to the activations and the straight-through-estimator to the training. Besides, we provide the detailed hyper-parameters in Appendix B. The code will be released for reproducibility.

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

# A  MORE EXPERIMENTS

In this section, we present more details about the experiments shown in Section 4. Table 3 shows the sparsity of each component of Q-Sparse and the baselines. Figure 7a and Figure 7b demonstrates that Q-Sparse achieve the similar convergence compared with the dense baseline, and can be used for the training of BitNet b1.58. Figure 7c presents the training loss curves of Q-Sparse and Block Q-Sparse.

We present the gradient's magnitude of each component for the dense baseline, Q-Sparse with and without STE estimator. As shown in Figure 8, STE estimator significantly eases the issue of gradient vanishing, especially at the bottom of the layers.

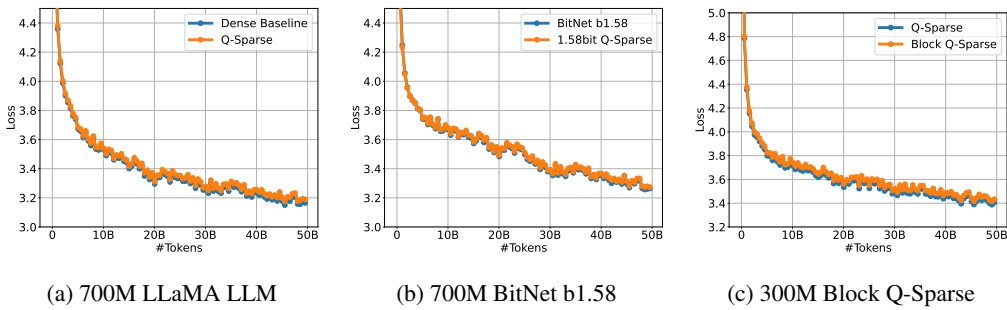

(a) 700M LLaMA LLM     (b) 700M BitNet b1.58     (c) 300M Block Q-Sparse

Figure 7: The training loss curve of Q-Sparse and the baseline with full-precision (a) and 1.58-bit (b) weight, Block Q-Sparse. We adopt top-$K$ as 70% for the experiments of BF16 and 1.58-bit weight, resulting in 40% overall sparsity. For the comparison with Block Q-Sparse, the sparsity ratio is 50% and the block size is set as 32.

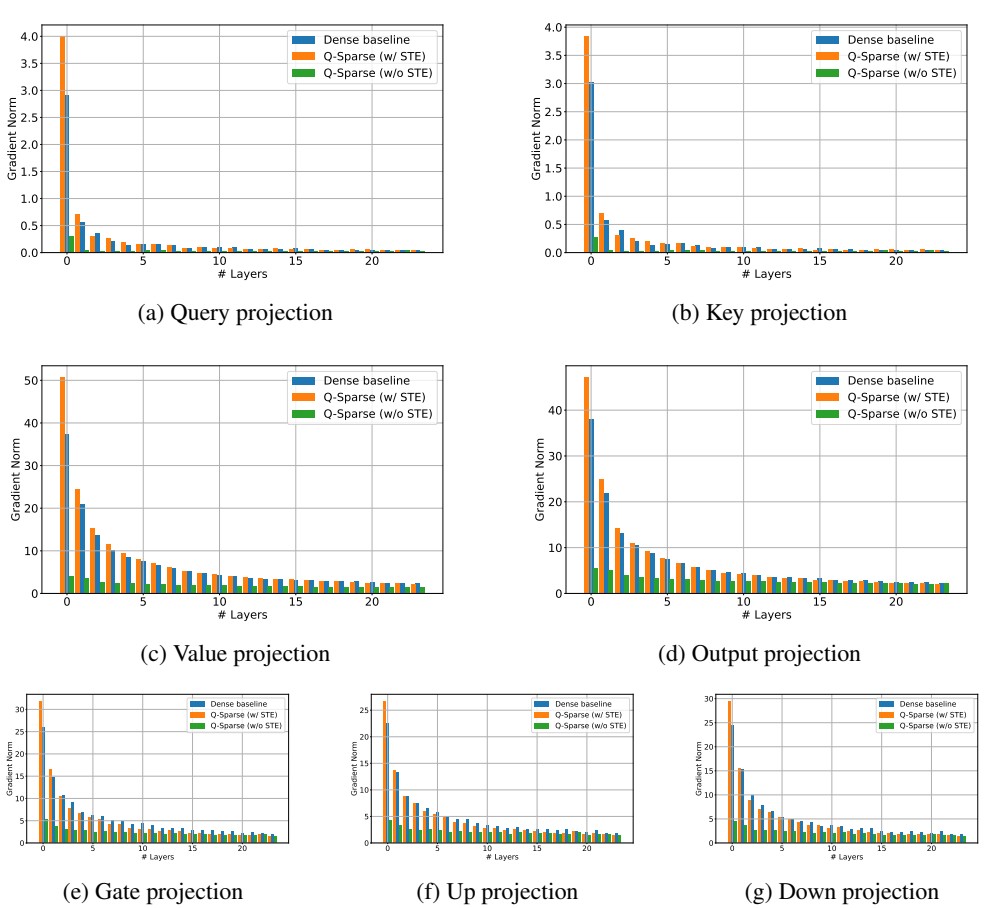

(a) Query projection     (b) Key projection

(c) Value projection     (d) Output projection

(e) Gate projection     (f) Up projection     (g) Down projection

Figure 8: The gradient magnitude of each linear projection of dense baseline, Q-Sparse with and without STE estimator across different layers.

## B  HYPERPARAMETERS

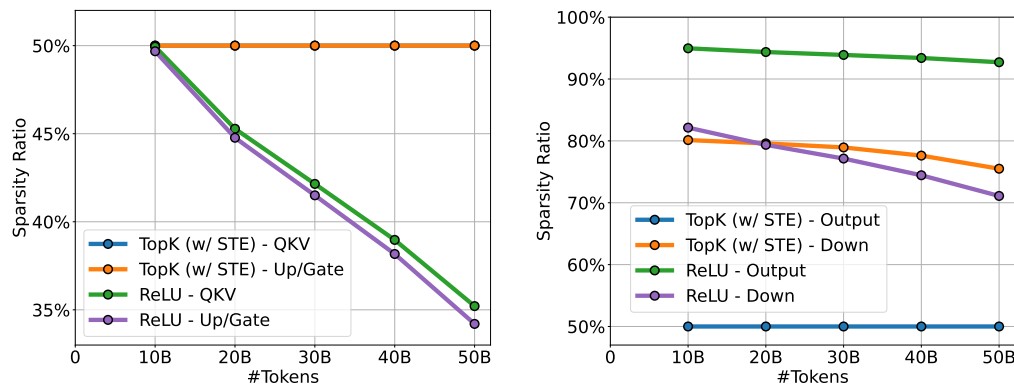

Figure 9: The sparsity ratio of each model's component of different sparsity functions.

| Models | Activated | QKV | Out | Up | Gate | Down | Overall |
|---|---|---|---|---|---|---|---|
| Dense Baseline | 7.0B | 0.0 | 0.0 | 0.0 | 0.0 | 0.0 | 0.0 |
| ReLUfication (Mirzadeh et al., 2023) | 5.0B | 12.3 | 0.0 | 10.3 | 10.3 | 79.3 | 28.3 |
| dReLU Sparsification (Song et al., 2024b) | 5.4B | 0.1 | 0.0 | 0.1 | 0.1 | 85.5 | 23.0 |
| **Q-Sparse (this work)** | 2.9B | 51.4 | 50.0 | 50.0 | 50.0 | 80.0 | 58.2 |
| | 3.8B | 42.0 | 40.0 | 40.0 | 40.0 | 60.4 | 45.7 |

Table 3: The activated parameters and the sparsity ratio of the continue-training for Q-Sparse and the baselines on the test set of Wikitext2.

| Size | Hidden Size | GLU Size | #Heads | #Layers | Seq Length |
|---|---|---|---|---|---|
| 300M | 1024 | 2730 | 16 | 24 | 2048 |
| 700M | 1536 | 4096 | 24 | 24 | 2048 |
| 1.3B | 2048 | 5460 | 32 | 24 | 2048 |
| 7B | 4096 | 11008 | 32 | 32 | 2048 |

Table 4: Model configurations for the scaling experiments of both BitNet b1.58 and LLaMA LLM with Q-Sparse.

| Model | Size | Learning Rate | Weight Decay | Batch Size | Adam $\beta$ |
|---|---|---|---|---|---|
| BitNet b1.58 | 300M | $1.8 \times 10^{-3} \to 1.5 \times 10^{-3}$ | $0.1 \to 0$ | 0.5M | (0.9, 0.95) |
| | 700M | $1.5 \times 10^{-3} \to 1 \times 10^{-3}$ | $0.1 \to 0$ | 0.5M | (0.9, 0.95) |
| | 1.3B | $1.2 \times 10^{-3} \to 8 \times 10^{-4}$ | $0.1 \to 0$ | 0.5M | (0.9, 0.95) |
| | 7B | $1 \times 10^{-3} \to 6 \times 10^{-4}$ | $0.1 \to 0$ | 0.5M | (0.9, 0.95) |
| LLaMA LLM | 300M | $6.0 \times 10^{-4}$ | 0.1 | 0.5M | (0.9, 0.95) |
| | 700M | $2.5 \times 10^{-4}$ | 0.1 | 0.5M | (0.9, 0.95) |
| | 1.3B | $2.0 \times 10^{-4}$ | 0.1 | 0.5M | (0.9, 0.95) |
| | 7B | $1.5 \times 10^{-4}$ | 0.1 | 0.5M | (0.9, 0.95) |

Table 5: Hyper-parameters for the scaling experiments of both BitNet b1.58 and LLaMA LLM with Q-Sparse.

| Hyperparameters | Value |
|---|---|
| Training updates | 10K |
| Tokens per sample | 4M |
| Adam $\beta$ | (0.9, 0.95) |
| Learning rate | 5e-5 |
| End learning rate | 1e-6 |
| Learning rate schedule | Polynomial decay |
| Warmup updates | 375 |
| Gradient clipping | 2.0 |
| Dropout | ✗ |
| Attention dropout | ✗ |
| Weight decay | 0.01 |

Table 6: Hyper-parameters for the continue-training of Mistral 7B with Q-Sparse on Findweb Edu dataset.

| Hyperparameters | Value |
|---|---|
| Training epoch | 1 |
| Batch Size | 128 |
| Adam $\beta$ | (0.9, 0.95) |
| Learning rate | {3e-6, 5e-6, 7e-6} |
| Learning rate schedule | Cosine decay |
| Warmup ratio | 0.03 |
| Dropout | ✗ |
| Attention dropout | ✗ |
| Weight decay | ✗ |

Table 7: Hyper-parameters for the supervised fine-tuning of Mistral 7B and Qwen-1.5 7B with Q-Sparse and Block Q-Sparse on OpenOrca dataset.

