# OpenReview forum: "Q-Sparse: All Large Language Models can be Fully Sparsely-Activated"
_ICLR.cc/2025/Conference — Submitted to ICLR 2025_

### Official Review · Reviewer_BiUz · 2024-10-30

**Soundness:** 3
**Presentation:** 3
**Contribution:** 3
**Rating:** 6
**Confidence:** 4

**Summary:**

The paper proposed Q-Sparse to enable full sparsity of activations in LLMs, aiming to improve the efficiency in inference. Different from prior works, Q-Sparse selectively activates the top-K features of each activation tensor. To stabilize the training, Q-Sparse also back-propagates the gradients of the non-activated features using straight-through estimator (STE). It was shown that Q-Sparse could be effectively applied to full-precision and 1-bit LLMs for pre-training, continued-training, and fine-tuning. The paper further presented the inference-optimal scaling law for sparsely activated LLMs, showing that they outperform their vanilla counterparts with the same inference compute budget (i.e. FLOPs).

**Strengths:**

i) Method. Q-Sparse is simple, yet effective. The proposed top-K sparsification could be implemented in modern compute units (e.g. CPU, GPU)

ii) Performance. Q-Sparse demonstrated superior performance compared to its vanilla counterparts with the same inference compute budget (i.e. FLOPs)

iii) Analysis. The paper provided the scaling law analysis for sparsely activated LLMs, which is potentially valuable to the community.

**Weaknesses:**

The reviewer has several concerns about the efficiency evaluation.

i) It was claimed Q-Sparse "can bring significant efficiency gains in inference" (L013), "including cost and energy consumption" (L023). However, in the current paper, these claims were only supported by the metrics like activated parameters or FLOPs, which is not fully convincing. Metrics like FLOPs do not have strong correlations with real-world efficiency like wall-clock speed-up, as argued in [1]. The authors are encouraged to demonstrate the inference efficiency of Q-Sparse by metrics that are directly linked to the inference overhead, e.g. speed, throughput, peak-memory usage, energy consumption, etc, as claimed in the abstract.


ii) The reviewer is also expecting some clarifications about the efficiency metrics used in the paper:

A) About the activated parameters. The reviewer agrees with the authors that the I/O for weight loading is one of the major bottlenecks for LLM inference (L038-040), as the weight matrices could be significantly larger than the batched activation tensors. The reviewer wonders how was the number of activated parameter calculated in the evaluation? Specifically, if the weight sharing for batched inputs was considered? This is important as for dense LLMs, the weight matrix may only be loaded once and reused for each input in the batch in GEMM. On the other hand, for sparsely activated LLMs, there were high chances that the full weight matrix may still be loaded if the activated channels, i.e. at least 40% for each input tensor, are uniformly distributed within the batch. In this case, the sparsely activated LLMs may show no I/O benefits compared to the dense LLMs.

B) About the FLOPs. Compared to the dense LLMs, Q-Sparse introduced extra operations during inference, i.e. dynamically computing the top-K activations for each tensor. The reviewer wonders if their computational overheads were also considered in the FLOP metric. The overheads of such operations could not be ignored as they typically have O(NlogK) complexity (i.e. K >= 40%*N) and cannot be parallelized directly (i.e. reduction).


[1] FlashAttention: Fast and Memory-Efficient Exact Attention with IO-Awareness. NeurIPS 2022.

**Questions:**

Please refer to Weaknesses

---

> ### Author Response · Authors · 2024-11-20
>
> We sincerely thank the reviewer for the insightful comments and questions, we would like to give detailed responses to your comments below.
>
> **Q1:** Inference speedup of Q-Sparse.
>
> **A1:** Thanks for your insightful question. We measure the inference speedup of Q-Sparse under different sparsity levels and the dense baseline. We benchmark the end-to-end single-batch decoding latency (token/s) by integrating it with the TEAL's kernel [1] on NVIDIA A6000.
>
> | Model            | Dense      | Sparsity 30%      | Sparsity 40%     | Sparsity 50%     | Sparsity 60%     |
> |------------------|---------|---------|---------|---------|--------|
> | Mistral-7B  | 43.890 (1.00x)  | 58.220 (1.33x)  | 62.920 (1.43x) | 71.590 (1.63x) | 80.55 (1.84x) |
> | Llama-2-7B    | 46.700 (1.00x) | 60.520 (1.30x) | 67.090 (1.44x) | 74.950 (1.60x) | 84.81 (1.82x) |
> | Llama-2-13B   | 24.180 (1.00x) | 33.480 (1.38x) | 37.530 (1.55x) | 42.450 (1.76x) | 48.74 (2.02x) |
> | Llama-2-70B   |  4.437 (1.00x) |  6.789 (1.53x) |  7.594 (1.71x) |  8.605 (1.94x) |  9.97 (2.25x) |
>
> As shown in the above table, Q-Sparse brings up to 1.71x, 1.94x, 2.25x speedup compared with the dense model when the sparsity is 40\% (TopK=60%), 50\% (TopK=50%) and 60\% (TopK=40%), respectively.
>
> **Q2:**  Inference speedup in batched inference.
>
> **A2:**
> Thanks for your valuable question.
> Q-Sparse focuses more on single-batch scenarios.
> For batched inference, we introduce Block Q-Sparse in Section 2.1, which leverages N:M sparsity, where N out of M consecutive elements to be zero, making it more suitable for large-batch inference with an optimized kernel.
> As presented in Figure 5(c) and Table 2, Block Q-Sparse achieves comparable performance to the dense baselines and Q-Sparse.
>
> The intrinsic balance property of N:M sparsity inherently ensures optimal and complete workload distribution during matrix partitioning for parallel computation.
> Furthermore, the N:M sparsity constraint on the sparse activations minimizes the locality of memory accesses to the dense weight, enhancing parallel thread efficiency on GPUs.
>
> We benchmark the latency (ms) of matrix multiplication with cuBLAS which is widely used for dense models, and the nmSPARSE-VW32 [2] on NVIDIA A100 GPU. The weight's shape is 1024 x 1024.
> | Batch size       | Dense      | Sparsity 50%      | Sparsity 75%     |
> |------------------|---------|---------|---------|
> | 1024  | 0.130 (1.00x)  | 0.094 (1.38x)  | 0.056 (2.30x) |
> | 4096  | 0.493 (1.00x)  | 0.311 (1.58x)  | 0.168 (2.94x) |
>
> As shown in above tables, nmSPARSE-VW32 achieves a speedup of 1.58x in 50% sparsity ratio and 2.94x in 75% sparsity ratio, when the weight's shape is 1024 x 1024 and the batch size is 4096.
>
> Since LLM decoding for half-precision models is memory-bound, quantization can be applied to reduce the time required to transfer model weights at each step, which is orthogonal to activation sparsity.
> Additionally, decoding with a 1.58-bit LLM is compute-bound rather than memory-bound, allowing it to be combined with Block Q-Sparse to further enhance inference performance.
>
> **Q3:** Whether the calculation of FLOPs considers Top-K.
>
> **A3:** We only consider the FLOPs of matrix multiplication, which is a major bottleneck for language models. For example, Given a linear projection $WX$, where $W$ has a shape of $d \times d$ and $X$ has a shape of $ d \times 1$, the FLOPs for the matrix multiplication are $ 2d^2$, while the FLOPs for the top-k operation are $O(d \log d)$.
>
> [1] Liu, J., Ponnusamy, P., Cai, T., Guo, H., Kim, Y., & Athiwaratkun, B. (2024). Training-free activation sparsity in large language models. arXiv preprint arXiv:2408.14690.
>
> [2] Lin, Bin, et al. "Efficient gpu kernels for n: M-sparse weights in deep learning." Proceedings of Machine Learning and Systems 5 (2023): 513-525.

---

> > ### Comment · Reviewer_BiUz · 2024-11-29
> > **Response to the rebuttal**
> >
> > The reviewer appreciates the rebuttal that addressed several of the initial concerns. The reviewer would like to keep the original positive score (i.e. 6)

---

### Official Review · Reviewer_cGCq · 2024-11-01

**Soundness:** 2
**Presentation:** 2
**Contribution:** 2
**Rating:** 5
**Confidence:** 3

**Summary:**

The authors analyzes the impact of sparsity on different components and experimental settings of LLM, and propose Q-Sparse to training sparsely activated large languge models.

**Strengths:**

The authors present valuable works on analyzing the impact of sparsity on LLM.

**Weaknesses:**

This paper lacks of methodological clarity in terms of novelty and the structure of the paper is not clear. Moreover, methodological novelty is limited since the proposed sparsity method may be considered as trick novelty in different components of LLM.

**Questions:**

1. The core principle of sparsity is not clear and is only explained by formulas, lacking a theoretical explanation.
2. The proposed sparsity method is more like a trick novelty since it is simple and applied to different components of LLM.
3. This paper has some findings and is more like an analytical paper, but what guiding significance do the experimental findings in the paper have? This can reflect the contribution of the paper to the LLM field.
4. It is recommended to provide comparative experiments with different models under different architectures on different datasets. The comparison in Table 1 is not sufficient to illustrate the effectiveness of the method.

---

> ### Author Response · Authors · 2024-11-20
>
> We sincerely thank the reviewer for the valuable feedback and questions, we would like to give detailed responses to your feedback below.
>
> **Q1:** Definition of activation sparsity.
>
> **A1:** Given a linear projection $WX$, where $W$ is the weight matrix and $X$ is the input, activation sparsity is defined as the proportion of zero entries in $X$. We will add the definition in the revised version of our paper.
>
> **Q2:** The guiding significance of experimental findings in the paper.
>
> **A2:** Thank you for your question. Q-Sparse enables fully sparse activation in LLMs, significantly improving inference efficiency. To the best of our knowledge, our work is the first to systematically investigate the quantitative relationship between the performance of LLM, activation sparsity level, and model size. We find that as model size increases, the performance gap between fully sparsely activated and dense LLMs narrows progressively. Furthermore, the relationship between the loss and model size follows a power-law, while the relationship between the loss and sparsity level follows an exponential law.
>
> Additionally, Q-Sparse is effective across pre-training, continued training and fine-tuning for off-the-shelf dense LLMs. As shown in Table 1 and Table 2, Q-Sparse has comparable performance to dense models while being more efficient at inference time. It can also be combined with the quantization (e.g., 1.58-bit LLMs) to further enhance the inference efficiency of LLMs.
>
> **Q3:** More comparative experiments with different models under different architectures on different datasets.
>
> **A3:** Thanks for your suggestion. Due to the large cost of continue-training, we compare Q-Sparse to ReLUfication and dReLU Sparsification in supervised fine-tuning. Following the setup shown in Section 4.3 of our paper, we finetune the base model of Mistral 7B and Qwen1.5 7B on Open-Orca dataset with ReLUfication and dReLU Sparsification.
>
> | Model            | Activated      | ARC      | HS      | MMLU     | WG     | TQA | Avg. |
> |------------------|---------|---------|---------|---------|--------|---------|--------|
> | Qwen1.5 7B  | 6.5B | 47.7 | 74.6 | 61.5 | 71.4 | 50.7 | 61.2|
> | ReLUfication | 4.8B | 30.3 | 25.6 | 24.5 | 49.8 | 50.1 | 36.0 |
> | dReLU Sparsification | 5.2B |42.0 | 57.6 | 39.0 | 64.1 | 47.1 | 50.0 |
> | Q-Sparse    | 3.6B | 46.3 | 72.6 | 59.1 | 67.5 | 50.3 | 59.2|
> | Q-Sparse    | 4.1B | 47.9 | 73.2 | 59.2 | 69.4 | 51.1 | 60.1|
>
> | Model            | Activated      | ARC      | HS      | MMLU     | WG     | TQA | Avg. |
> |------------------|---------|---------|---------|---------|--------|---------|--------|
> | Mistral 7B  | 7.0B | 62.5 | 82.6 | 61.2 | 77.6 | 50.3 | 66.8 |
> | ReLUfication | 5.0B | 35.8 | 47.5 | 26.0 | 54.9 | 42.1 | 41.3 |
> | dReLU Sparsification | 5.4B | 31.9 | 45.1 | 32.0 | 54.1 | 45.2 | 41.6 |
> | Q-Sparse    | 3.8B | 60.5 | 81.5 | 60.0 | 77.1 | 50.5 | 65.9 |
> | Q-Sparse    | 4.3B | 61.4 | 81.6 | 60.6 | 77.6 | 50.7 | 66.4 |
>
> As shown in the above tables, Q-Sparse significantly outperforms ReLUfication and dReLU Sparsification in terms of both sparsity and performance. It indicates that ReLUfication and dReLU Sparsification require large amount of training data to make the model adapt to sparse activations, thus are not suitable for fine-tuning.

---

> > ### Comment · Reviewer_cGCq · 2024-12-02
> > **Reply**
> >
> > Thanks for the author's reply, I am currently inclined to maintain this score and will need more time to check the details.

---

### Official Review · Reviewer_4NHE · 2024-11-04

**Soundness:** 2
**Presentation:** 2
**Contribution:** 2
**Rating:** 3
**Confidence:** 4

**Summary:**

The authors introduced Q-Sparse that enables full sparsity of activations for efficient inference. They also presented an inference-optimal scaling law for sparsely-activated LLMs. They conducted experiments on commonsense reasoning tasks and MMLU.

**Strengths:**

1. The sparsely-activated models with around 40% sparsity ratio can perform comparably to the dense baselines with the same model size and training tokens.

2. The performance gap between the sparsely-activated models and the dense baselines decreases as the number of parameters goes up.

**Weaknesses:**

1. The paper mentioned "Q-Sparse enables full sparsity of activations in LLMs which can bring significant efficiency gains in inference", but there is neither discussion nor experiment about measuring the execution time and/or peak GPU memory usage.

2. The evaluation for math (e.g., GSM8K, MATH) and code (e.g., HumanEval, MBPP) would be required to ensure that the performance of sparsely-activated models can match the performance of the dense baselines with the same model size and training tokens. Given that  Orca’s capability is designed for open-ended generation, only the evaluation for commonsense reasoning tasks and MMLU seems not enough.

**Questions:**

How about conducting experiments for Llama models (e.g., Llama 3 8B)? The reason why I brining it up is that Llama is more popular than Mistral and Qwen, so I believe that using Llama is more beneficial to researchers and engineers.

---

> ### Author Response · Authors · 2024-11-20
>
> We would like to thank you for your time and constructive suggestions, we would like to give detailed responses to your comments below.
>
> **Q1:** Inference speedup of Q-Sparse.
>
> **A1:** Thanks for your constructive question. We further conduct experiments to measure the inference speedup of Q-Sparse under different sparsity level and the dense baseline. We benchmark the end-to-end single-batch decoding latency (token/s) by integrating it with TEAL's kernel [1] on NVIDIA A6000.
>
> | Model            | Dense      | Sparsity 30%      | Sparsity 40%     | Sparsity 50%     | Sparsity 60%     |
> |------------------|---------|---------|---------|---------|--------|
> | Mistral-7B  | 43.890 (1.00x)  | 58.220 (1.33x)  | 62.920 (1.43x) | 71.590 (1.63x) | 80.55 (1.84x) |
> | Llama-2-7B    | 46.700 (1.00x) | 60.520 (1.30x) | 67.090 (1.44x) | 74.950 (1.60x) | 84.81 (1.82x) |
> | Llama-2-13B   | 24.180 (1.00x) | 33.480 (1.38x) | 37.530 (1.55x) | 42.450 (1.76x) | 48.74 (2.02x) |
> | Llama-2-70B   |  4.437 (1.00x) |  6.789 (1.53x) |  7.594 (1.71x) |  8.605 (1.94x) |  9.97 (2.25x) |
>
> As shown in the above table, Q-Sparse brings up to 1.71x, 1.94x, 2.25x speedup compared with the dense model when the sparsity is 40\% (TopK=60%), 50\% (TopK=50%) and 60\% (TopK=40%), respectively.
>
> **Q2:**  Add evaluation for open-ended generation (e.g., Math or Code).
>
> **A2:** Thanks for your question. We finetune the base model of Mistral 7B and Qwen1.5 7B on MetaMath dataset [2]. The batch size is set as 128. The learning rates are selected from {3e-6, 5e-6}. All models are trained with 1 epoch for a fair comparison. We evaluate 5-shot GSM-8K using LM Harness.
>
> | Model            |  Activated | GSM-8K  |
> |------------------|---------|---------|
> | Qwen-1.5 7B      |  6.5B  |  68.84  |
> | Q-Sparse         |  3.6B  |  67.78  |
> | Q-Sparse         |  4.1B  |  69.60  |
>
> | Model            |  Activated | GSM-8K  |
> |------------------|---------|---------|
> | Mistral 7B       |  7.0B   |  68.01  |
> | Q-Sparse         |  3.8B   |  67.55  |
> | Q-Sparse         |  4.3B   |  69.45  |
>
> As shown in the above tables, Q-Sparse can be used to finetune a dense pretrained model to a sparse model with almost no loss at accuracy on GSM-8K dataset.
>
> [1] Liu, J., Ponnusamy, P., Cai, T., Guo, H., Kim, Y., & Athiwaratkun, B. (2024). Training-free activation sparsity in large language models. arXiv preprint arXiv:2408.14690.
>
> [2] Yu, L., Jiang, W., Shi, H., Yu, J., Liu, Z., Zhang, Y., ... & Liu, W. (2023). Metamath: Bootstrap your own mathematical questions for large language models. arXiv preprint arXiv:2309.12284.

---

> > ### Comment · Reviewer_4NHE · 2024-11-30
> >
> > (1) I believe that the inference speed-up of Q-Sparse should be analyzed with diverse hardwares (e.g., H100, A100, L40), not only one hardware, to argue that Q-Sparse enables full sparsity of activations in LLMs which can bring significant efficiency gains in inference.
> >
> > (2) I cannot understand why the authors fine-tuned the base model of Mistral 7B and Qwen1.5 7B on MetaMath dataset with only one epoch. In the MetaMath paper, three epochs are recommended.
> >
> > (3) There is no response to my question.
> >
> > Accordingly, I keep my original score.

---

### Official Review · Reviewer_En89 · 2024-11-05

**Soundness:** 3
**Presentation:** 3
**Contribution:** 2
**Rating:** 5
**Confidence:** 3

**Summary:**

This paper proposes a method to train sparsely-activated large language models (LLMs) that are more efficient at inference. The authors introduce the use of squared ReLU to enhance sparsity and apply STE to address the gradient vanishing problem during the training of these sparsely-activated LLMs. Experiments are conducted on both dense and quantized models.

**Strengths:**

1. The paper is well-written and easy to follow.
2. The extension of the scaling law to sparsely-activated LLMs in Section 3 is well-motivated and innovative.
3. Experimental evaluations are comprehensive, including both dense and quantized LLMs, reflecting realistic deployment settings.

**Weaknesses:**

1. MoE models are also a type of sparsely-activated model. It would be valuable to include a comparison between MoE models and the proposed sparsely-activated models in terms of accuracy and efficiency.
2. A concern lies in the actual inference speedup of the proposed method, especially during the decoding step in batched inference. In batched settings, the columns activated may vary across cases, and in the worst case, it may still be necessary to load all weights during each decoding step. Since the decoding step can be memory-bound rather than compute-bound at certain batch sizes, this could potentially limit the speedup achieved by the proposed method in practice.
3. While empirical results suggest that STE is effective, it remains unclear why gradients should be maintained for non-activated neurons. A more rigorous theoretical discussion or proof would strengthen this point.
Questions:

**Questions:**

Given that MoE models are a special type of sparsely-activated model, what are the connections and differences between the scaling law for MoE models and the scaling law proposed here?

---

> ### Author Response · Authors · 2024-11-20
>
> We thank the reviewer for the valuable comments and questions, we would like to give detailed responses to your comments below.
>
> **Q1:** Comparison between MoE and Q-Sparse.
>
> **A1:** Our approach is orthogonal to MoE and can be combined with it to further improve inference efficiency. MoE models maintain a large expert pool, where only a small number of experts are activated for each token. Q-Sparse extends this sparsity to every linear projection. During matrix multiplication, Q-Sparse prunes entries with smaller magnitudes for each token, thereby reducing the FLOPs during inference.
>
> **Q2:**  Inference speedup in batched inference.
>
> **A2:** Thanks for your valuable question.
> Q-Sparse focuses more on single-batch scenarios.
> For batched inference, we introduce Block Q-Sparse in Section 2.1, which leverages N:M sparsity, where N out of M consecutive elements to be zero, making it more suitable for large-batch inference with an optimized kernel.
> As presented in Figure 5(c) and Table 2, Block Q-Sparse achieves comparable performance to the dense baselines and Q-Sparse.
>
> The intrinsic balance property of N:M sparsity inherently ensures optimal and complete workload distribution during matrix partitioning for parallel computation.
> Furthermore, the N:M sparsity constraint on the sparse activations minimizes the locality of memory accesses to the dense weight, enhancing parallel thread efficiency on GPUs.
>
> We benchmark the latency (ms) of matrix multiplication with cuBLAS which is widely used for dense models, and the nmSPARSE-VW32 [1] on NVIDIA A100 GPU. The weight's shape is 1024 x 1024.
> | Batch size       | Dense      | Sparsity 50%      | Sparsity 75%     |
> |------------------|---------|---------|---------|
> | 1024  | 0.130 (1.00x)  | 0.094 (1.38x)  | 0.056 (2.30x) |
> | 4096  | 0.493 (1.00x)  | 0.311 (1.58x)  | 0.168 (2.94x) |
>
> As shown in above tables, nmSPARSE-VW32 achieves a speedup of 1.58x in 50% sparsity ratio and 2.94x in 75% sparsity ratio, when the weight's shape is 1024 x 1024 and the batch size is 4096.
>
> Since LLM decoding for half-precision models is memory-bound, quantization can be applied to reduce the time required to transfer model weights at each step, which is orthogonal to activation sparsity.
> Additionally, decoding with a 1.58-bit LLM is compute-bound rather than memory-bound, allowing it to be combined with Block Q-Sparse to further enhance inference performance.
>
> **Q3:**  Discussion about why gradients should be maintained for non-activated neurons
>
> **A3:** We presents the gradients and loss curves of Q-Sparse with and without STE in Figure 2 and Figure 6, respectively. Adding STE significantly alleviates the issue of smaller gradients in the lower layers caused by activation sparsity, thus accelerating the model's convergence.
>
> [1] Lin, Bin, et al. "Efficient gpu kernels for n: M-sparse weights in deep learning." Proceedings of Machine Learning and Systems 5 (2023): 513-525.

---

### Author Response · Authors · 2024-11-24
**Looking forward to More discussion**

Dear reviewers,

Thanks for your meticulous review and valuable time, which played a pivotal role in enhancing the quality of our paper.
We greatly appreciate your acknowledgment of the strengths in our work, particularly regarding the analysis of scaling laws for sparsely-activated LLM, comprehensive experimental evaluations. We apologize for any inconvenience, but as the deadline is approaching. We would like to provide an update on our progress.

In our rebuttal, we have addressed your concerns in detail, including issues related to speedup in batched inference, decoding latency of different sparsity levels, experiments on math tasks and comparison with baselines on fine-tuning.

We hope that these clarifications resolve your confusion, and we will improve our paper on your feedback and suggestions. Please let us know if there are any additional questions or clarifications needed.

We sincerely appreciate your time and effort in reviewing our paper, and we would greatly appreciate it if you could re-consider the rating.


Best regards,

Authors.

---

### Meta-Review · Area_Chair_kjPx · 2024-12-13

**Metareview:**

While the work introduces an interesting approach for sparse activation in LLMs and explores efficiency gains, it lacks sufficient clarity and rigor in several key areas. The methodology, while claiming innovation, appears incremental, building primarily on existing sparse activation techniques. The paper also fails to provide adequate theoretical justification for critical components like the straight-through estimator (STE) and its impact on gradient flow. Without rigorous analysis, these claims remain speculative, diminishing the paper's contribution. Additionally, the experimental results lack breadth, with insufficient comparisons to established baselines like Mixture of Experts (MoE) models and limited evaluation on complex tasks.

**Additional Comments On Reviewer Discussion:**

During the rebuttal, reviewers raised several points: (1) lack of empirical evidence for real-world efficiency gains such as speed, memory, or energy improvements beyond FLOP reductions; (2) insufficient theoretical justification for the use of STE and its impact on sparsity training; (3) limited experimental evaluation, particularly comparisons with MoE models and performance on diverse datasets like math or code tasks; and (4) clarity concerns regarding methodological novelty and guiding significance of findings. The authors partially addressed these concerns by providing additional benchmarks on decoding latency, more datasets for evaluation, and clarifications on sparsity definitions and comparisons. However, key concerns about theoretical rigor, comprehensive efficiency metrics, and broader evaluation were not fully resolved.

---

### Decision · Program_Chairs · 2025-01-22

Reject